# CoLaSplat: Compact Language 3D Gaussian Splatting

## Abstract

Language 3D Gaussian Splatting (3DGS) has exhibited promising advancements in open-vocabulary 3D scene understanding, incorporating semantic features from pretrained vision-language models into Gaussians to encode the semantic information of a scene. However, language-embedded 3DGS suffers from high computational and storage costs due to the massive number of Gaussians and the extra high-dimensional semantic attributes, which hinder its practical application. Existing compression methods primarily reduce 3DGS model redundancy through pruning or quantization, which can be sequentially applied to obtain a highly compressed language-embedded 3DGS model as a straightforward solution. However, all the existing approaches are not designed for compressing language 3DGS, where rich semantic features are ignored during the compression stages, leading to severe semantic information loss and significantly degraded scene understanding performance. Furthermore, the disjoint nature of the pruning and quantization stages results in lower rendering quality. To address these issues, we propose **CoLaSplat**, a unified compression framework for compact language 3DGS. **CoLaSplat** formulates semantic learning, sparsification, and vector quantization as a single optimization problem, constrained by the number of Gaussian primitives and vector quantization objective, seamlessly integrating the optimization procedure into the training process and incorporating language embeddings. To solve the unified optimization problem, we develop an efficient primal-dual optimization scheme by solving their associated subproblems and updating the variables separately, progressively compacting the model while preserving semantic and RGB rendering fidelity. Moreover, we theoretically analyze the convergence and stability of the proposed framework. Extensive experiments on 3D semantic segmentation and object localization demonstrate that our proposed **CoLaSplat** brings substantial efficiency gains while maintaining high task performance. **Specifically, CoLaSplat achieves up to $15\times$ model size reduction, $147\times$ faster inference, and $6.7\times$ lower memory usage.**

## 1 Introduction

Open-vocabulary 3D scene understanding has received substantial attention in the field of artificial intelligence. It aims to comprehend and interpret 3D scenes with natural language, facilitating a wide range of applications, such as immersive AR/VR experiences (Koch et al., 2024), autonomous driving (Cheng & Li, 2024), and robotic manipulation (Qiu et al., 2024a; Huang et al., 2022). Prior works primarily rely on implicit neural representations (Peng et al., 2023; Wang et al., 2023; Kerr et al., 2023) to capture 3D representations. Recently, 3D Gaussian Splatting (3DGS) (Kerbl et al., 2023) has revolutionized the realm of 3D scene representation learning. Instead of implicit representations, it leverages explicit point-based representations learned by millions of 3D Gaussians to model 3D scene geometric and appearance details, achieving superior visual fidelity and real-time rendering. Inspired by the promising visual rendering results of 3DGS, current 3D scene understanding approaches (Zhou et al., 2024; Qiu et al., 2024b; Qu et al., 2024) have shifted to develop *language-embedded* 3DGS by enriching each Gaussian with semantic features, which are extracted from the pre-trained vision-language model such as CLIP (Radford et al., 2021) and BLIP (Li et al., 2022b), thereby endowing it with 3D semantic representation capability.

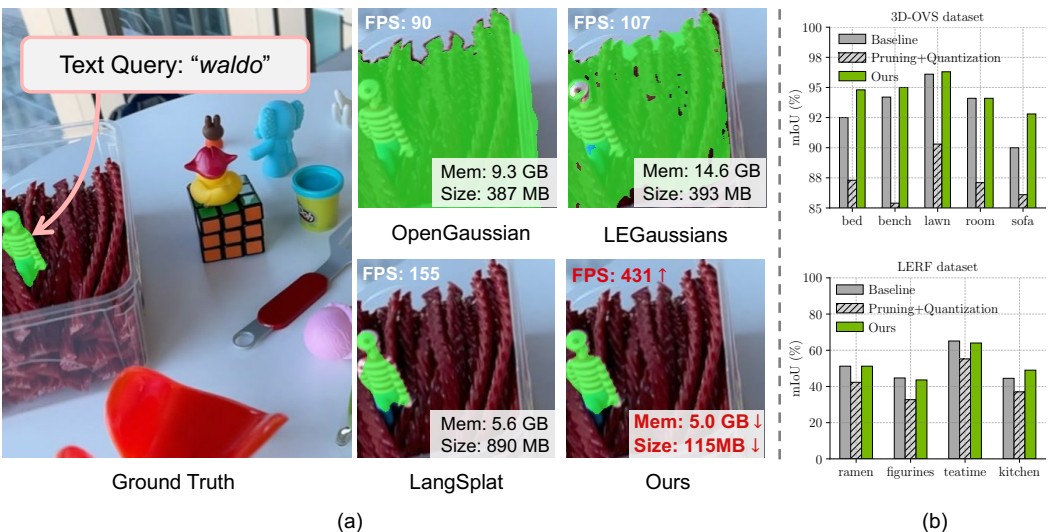

Figure 1: (a) Improvements of **CoLaSplat** over baselines in 3D semantic segmentation and model efficiency on the "*figurines*" scene from LERF (Kerr et al., 2023) dataset. (b) 3D semantic segmentation performance (mIoU, %) on 3D-OVS (Liu et al., 2023) dataset and LERF dataset, comparing the baseline (Langsplat), a simple pruning (Zhang et al., 2025b) and quantization (Navaneet et al., 2023) combination, and **CoLaSplat**.

Despite their strengths, existing language 3DGS methods suffer from significant *memory and storage challenges*, which mainly arise from two aspects. First, inheriting the substantial number of Gaussians associated with trainable parameters (e.g., opacity, location, and color) (Kerbl et al., 2023) from 3DGS, language 3DGS requires massive memory space to store Gaussians. Moreover, the high-dimensional semantic features that are embedded into Gaussian primitives have further significantly increased their memory consumption (Zhou et al., 2024), especially in densely sampled scenes, thereby preventing prior approaches from semantically understanding complex 3D scenes.

Unfortunately, previous works have focused on compressing standard 3DGS, with pruning (Yang et al., 2024; Ali et al., 2024; Zhang et al., 2025b) and quantization (Navaneet et al., 2023; Liu et al., 2024; Lee et al., 2024). To obtain a highly compact language-embedded 3DGS model, a naive solution is to sequentially apply existing pruning and quantization methods, as they address orthogonal sources of redundancy (Hanson et al., 2025; Navaneet et al., 2024; Fan et al., 2024). However, since these approaches are not specifically designed for language 3DGS, simply applying them can result in severe semantic information loss, thereby degrading scene understanding performance, as illustrated in Fig. 1(b). Additionally, the pruning and quantization compression stages are disjoint, which can accumulate and amplify errors, leading to unsatisfactory rendering quality.

To address these issues, we propose **CoLaSplat**, a unified compression framework for compact and high-fidelity language-embedded 3DGS. **CoLaSplat** innovatively unifies model training, pruning, and vector quantization as a single optimization problem, constrained by the number of Gaussian primitives and the vector quantization objective, seamlessly integrating the optimization procedure into the training process, which automatically finds the sweet spot among multiple objectives. To solve this non-trivial optimization problem, we develop a primal-dual optimization scheme that connects Gaussian parameters with an auxiliary variable and the set of quantized parameter vectors. Then, multiple iterative steps are alternatively performed in the optimization-integrated training until convergence. This process progressively removes unimportant Gaussians and quantizes parameters of Gaussians while maximally preserving semantic and color information. This enables **CoLaSplat** to substantially reduce both the number of Gaussian primitives and parameter redundancy in language 3DGS, considerably improving computational efficiency while maintaining semantic and visual fidelity.

In summary, the main contributions of our work can be summarized as:

- We propose **CoLaSplat**, a unified language 3DGS compression framework that alleviates memory and storage costs. By formulating the compression and semantic learning objective as a unified optimization problem and iteratively solving it, **CoLaSplat** progressively sparsifies Gaussian primitives and quantizes the parameters in the training process while preserving semantic and rendering fidelity. Thus, **CoLaSplat** significantly reduces the model size while maintaining high-quality semantic representations. To the best of our knowledge, **CoLaSplat** is the first unified compression framework for compact language 3DGS, enabling accurate open-vocabulary scene understanding and high-quality rendering with highly reduced computational costs.

- We propose an efficient primal-dual optimization solution to solve the unified compression problem, which alternates among four steps: optimizing the supervision loss with a regularization term through a *primal update*, enforcing sparsity through a *sparsification update*, imposing quantization objective through a *vector quantization update*, and *dual update*. Moreover, we provide a rigorous convergence analysis and proof of our method in Appendix C.

- We conduct extensive experiments to evaluate the effectiveness of **CoLaSplat** on multiple 3D open-vocabulary understanding tasks, including 3D semantic segmentation and 3D object localization. Particularly, it achieves significant reductions in model size of up to $15\times$, GPU memory usage of up to $6.7\times$, and speedup in inference of up to $147\times$, while maintaining superior semantic and visual quality comparable to the state-of-the-art baselines.

## 2 RELATED WORK

**Open-Vocabulary 3D Scene Understanding.** Previous methods (Li et al., 2022a; Liang et al., 2023; Kerr et al., 2023) primarily leverage implicit neural networks operating on 2D images, using vision-language models like CLIP (Radford et al., 2021) to achieve cross-modal feature alignment. Their core objective is to overcome the limitations of traditional segmentation methods that rely on predefined categories. Recent efforts mainly concentrate on 3DGS-based implementations. In these approaches, each Gaussian is augmented with semantic feature embeddings. Guo et al. (2024) enables semantic-based selection and editing of Gaussians; Wu et al. (2024) incorporates a CLIP text encoder to align Gaussian semantics with textual queries; Qin et al. (2024) assigns category labels through Gaussian-text feature matching, addressing the need for 3D annotations in conventional 3D segmentation; Zhou et al. (2024) employs a feed-forward 3D Gaussian architecture with a sparse view feature inference module, eliminating the need for per-scene optimization; and Shi et al. (2024) adds an edge detection module on top of CLIP-based semantic alignment to resolve semantic ambiguities around object boundaries in open-vocabulary 3D segmentation.

**3D Gaussian Splatting Compression.** Techniques for compressing 3D Gaussian Splatting are commonly divided into three categories: pruning, quantization, and mixed compression. Pruning techniques focus on discarding unimportant Gaussians through the learnable mask (Lee et al., 2024; Wang et al., 2024; Liu et al., 2025; Zhang et al., 2025b; 2024; 2025a; Fang & Wang, 2024), view-dependent metrics (Fan et al., 2024) or hand-crafted importance criteria such as opacity, composited importance score, and dominant primitives (Niemeyer et al., 2024; Ali et al., 2024; Hanson et al., 2025; Fan et al., 2024). Quantization-based efforts aim to reduce the number of values to represent each parameter in Gaussians (Navaneet et al., 2024). Leveraging the size estimator to establish a robust relationship between size and hyperparameters, SizeGS (Xie et al., 2024) proposes a size-aware hierarchical mixed precision quantization scheme. To achieve higher compression rates, more recent approaches combine multiple compression techniques for mixed compression (Niedermayr et al., 2024; Deng et al., 2024). SA-3DGS (Zhang et al., 2025a) removes the least significant Gaussians based on learned importance scores and further compresses their parameters via importance-aware clustering. CompGS (Liu et al., 2024) employs sliding-window masking and geometry-based quantization to compress redundant Gaussians and their geometric attributes.

However, these methods are designed to eliminate redundancy in standard 3DGS, whereas the compression of language-embedded 3DGS has not yet been explored.

## 3 BACKGROUND

3D Gaussian Splatting explicitly leverages a collection of Gaussians to model 3D scene geometry and appearance. Specifically, each Gaussian $G(\boldsymbol{x})$ is defined by a mean vector $\boldsymbol{\mu} \in \mathbb{R}^3$ and a

covariance matrix $\boldsymbol{\Sigma}$:

$$G(\boldsymbol{x}) = \exp\left(-\frac{1}{2}(\boldsymbol{x} - \boldsymbol{\mu})^\top \boldsymbol{\Sigma}^{-1}(\boldsymbol{x} - \boldsymbol{\mu})\right), \tag{1}$$

where $\boldsymbol{x}$ is a location in the 3D scene. The learnable parameters of a Gaussian consist of $\{\boldsymbol{\mu}, \boldsymbol{c}, o, \boldsymbol{R}, \boldsymbol{S}\}$, which correspond to its position, color, opacity, rotation, and scale.

To optimize the Gaussian parameters, they are projected onto 2D image planes, and a tile-based rasterization strategy is employed to improve computational efficiency (Zwicker et al., 2001). The 2D image pixel color $\boldsymbol{C}_{\text{pixel}}$ is rendered by the blending process:

$$\boldsymbol{C}_{\text{pixel}} = \sum_{i \in \mathcal{N}} \boldsymbol{c}_i\, \alpha_i \prod_{j=1}^{i-1}(1 - \alpha_j), \quad \alpha_i = o_i\, G_i', \tag{2}$$

where $\boldsymbol{c}_i$ is the color of the $i$-th Gaussian, $o_i$ is its opacity, $\mathcal{N}$ is the set of ordered Gaussians contributing to the rasterization at the target rendering pixel, and $G_i'$ represents the projection of the Gaussian onto the 2D plane.

To embed semantic information into Gaussian primitives, learnable language embeddings $\boldsymbol{f}$ are introduced as new attributes associated with each Gaussian. For a particular pixel, the rasterization process propagates these embeddings onto the image plane as follows:

$$\boldsymbol{F}_{\text{pixel}} = \sum_{i \in \mathcal{N}} \boldsymbol{f}_i\, \alpha_i \prod_{j=1}^{i-1}(1 - \alpha_j), \tag{3}$$

where $\boldsymbol{F}_{\text{pixel}}$ represents the accumulated language embedding related to a pixel in a 2D image.

Some recent advanced methods (Nguyen et al., 2024; Qin et al., 2024) embed multi-scale hierarchical language features, derived from CLIP (Radford et al., 2021) features and guided by multi-scale masks obtained from Segment Anything Model (SAM) (Kirillov et al., 2023), typically covering three semantic scales. Moreover, dimensionality reduction of language embeddings is often necessary; for instance, LangSplat (Qin et al., 2024) employs an autoencoder to map CLIP embeddings into a compact latent space for efficient rendering and reconstructs the full features when needed. Our approach also adopts both strategies.

During inference, given a text query $\phi_{\text{query}}$, a relevancy score (Kerr et al., 2023) is computed between $\phi_{\text{query}}$ and the rendered language embedding $\boldsymbol{F}_{\text{pixel}}$ for downstream tasks. For 3D object localization, the point with the highest relevancy score is predicted as the object location and considered correct if inside the ground-truth bounding box. For 3D semantic segmentation, points exceeding a relevancy threshold are assigned the query category to form segmentation masks.

## 4 METHODOLOGY: **CoLaSplat**

To obtain a highly compact language-embedded 3DGS model with the maximum preserved semantic information and high-quality rendering, we propose a unified language 3DGS compression framework, **CoLaSplat**, which tackles this complex multi-objective compression by formulating the entire process as a single optimization with constraints on the number of Gaussian primitives and the vector quantization loss. Then, we propose an efficient primal-dual solution to optimize the 3DGS language model, which alternates among optimizing multiple sub-problems with iterative update steps.

### 4.1 PROBLEM FORMULATION

**Language Embedded Training Objective.** In the regular 3DGS training (Kerbl et al., 2023), the Gaussian parameters are learned by a combination of pixel-wise $\ell_1$ loss and differentiable SSIM loss between the rendered RGB images and the ground-truth multi-view images. The rendering loss is defined as $L_{\text{RGB}} = (1 - \lambda)L_1 + \lambda L_{\text{D-SSIM}}$, where $\lambda \in [0, 1]$ balances the two terms. On the other side, the language embeddings are trained with multi-level semantic labels in a supervised way. As introduced in the previous section, the ground truths are derived from CLIP features and structured under the guidance of multi-scale masks generated by SAM. Mathematically, the language loss is

defined as $L_{\text{lang}} = \|\boldsymbol{F} - \boldsymbol{F}_{\text{GT}}\|_1$, where $\boldsymbol{F}_{\text{GT}}$ is the ground truth language embeddings. These two losses are combined to form the final supervision loss for optimizing the given scene:

$$L = \underbrace{(1 - \lambda)L_1 + \lambda L_{\text{D-SSIM}}}_{\text{RGB learning}} + \underbrace{\gamma\|\boldsymbol{F} - \boldsymbol{F}_{\text{GT}}\|_1}_{\text{semantic alignment}}, \tag{4}$$

where $\gamma$ denotes the weighting factor that balances the rendering and language losses. Minimizing this loss yields a language 3DGS model that enhances visual fidelity while embedding semantic information.

**Unified Optimization Objective with Sparsity and Vector Quantization Constraints.** Recalling the rendering functions in Eq. 2 and Eq. 3, the contribution of each Gaussian primitive to the rendered color and semantic results is positively correlated with its opacity. Therefore, we can constrain the number of Gaussians that contribute significantly to the rendering results, effectively sparsifying the model. Quantization is then applied only to the remaining parameters.

Given a 3DGS model with $N$ initial Gaussians, we represent the opacities of all Gaussians as a vector $\boldsymbol{o} = [o_1, o_2, \ldots, o_N] \in \mathbb{R}^N$, where $o_i$ denotes the opacity of the $i$-th Gaussian, and the remaining parameters as $\boldsymbol{\Theta} = \{\boldsymbol{\theta}_1, \boldsymbol{\theta}_2, \ldots, \boldsymbol{\theta}_N\}$, which include all parameters other than opacity. The training process is then formulated with a loss function $L(\boldsymbol{o}, \boldsymbol{\Theta})$ that depends on both the opacities $\boldsymbol{o}$ and the other parameters $\boldsymbol{\Theta}$. We denote the set of clusters as $\mathcal{Q} = \{Q_1, Q_2, \ldots, Q_M\}$, where each cluster $Q_j$ has a centroid $\boldsymbol{q}_j$. The collection of centroids $\{\boldsymbol{q}_1, \boldsymbol{q}_2, \ldots, \boldsymbol{q}_M\}$ constitutes the quantization codebook, where $M$ is the number of cluster centers, i.e., the codebook size. The centroids are updated by clustering the set of parameters vectors $\boldsymbol{\Theta}$ and are also the quantization vectors stored in the codebook. With the clusters and codebook defined, the unified optimization objective, which includes sparsity and vector quantization constraints, is given by:

$$\min_{\boldsymbol{o}, \boldsymbol{\Theta}, \mathcal{Q}} L(\boldsymbol{o}, \boldsymbol{\Theta}) + \sum_{j=1}^{M} \sum_{\boldsymbol{\theta}_i \in Q_j} \|\boldsymbol{\theta}_i - \boldsymbol{q}_j\|_2^2, \quad \text{s.t.} \quad \mathbf{card}(\boldsymbol{o}) \leq \kappa. \tag{5}$$

The *sparsity constraint* $\mathbf{card}(\boldsymbol{o}) \leq \kappa$ enforces sparsity in the Gaussian representation by limiting the number of Gaussians with non-zero opacity to at most $\kappa$. Here, $\mathbf{card}(\boldsymbol{o})$ denotes the *cardinality* of the vector $\boldsymbol{o}$, i.e., the number of its non-zero elements. During training, this encourages information to concentrate on a small subset of Gaussians with high opacity, while the others gradually become transparent. After convergence, the nearly transparent Gaussians can be discarded, yielding a compact representation.

The *quantization penalty* $\sum_{j=1}^{M} \sum_{\boldsymbol{\theta}_i \in Q_j} \|\boldsymbol{\theta}_i - \boldsymbol{q}_j\|_2^2$ acts as the vector quantization objective, penalizing deviations of each parameter vector $\boldsymbol{\theta}_i$ from its assigned cluster centroid $\boldsymbol{q}_j$. Here, $\boldsymbol{\theta}_i \in Q_j$ denotes all vectors belonging to cluster $Q_j$. This term encourages each vector to remain close to its centroid, effectively promoting quantization of the parameter space. After training, every vector can be approximated by its cluster center, requiring only the index of the center and the codebook $\{\boldsymbol{q}_1, \ldots, \boldsymbol{q}_M\}$ to be stored. Since $M \ll N$, this significantly reduces storage cost compared to storing all individual vectors.

## 4.2 Optimization

The unified optimization objective in Eq. 5 involves two non-differentiable components: the *sparsity constraint* $\mathbf{card}(\boldsymbol{o}) \leq \kappa$ and the *quantization penalty* $\sum_{j=1}^{M} \sum_{\boldsymbol{\theta}_i \in Q_j} \|\boldsymbol{\theta}_i - \boldsymbol{q}_j\|_2^2$, where the penalty itself is differentiable but the discrete assignment $\boldsymbol{\theta}_i \in Q_j$ is not. To handle the non-differentiable components that make the optimization challenging, we reformulate both constraints into forms that are more amenable to optimization.

We first introduce an auxiliary variable $\boldsymbol{y}$ to separate the sparsity constraint from $\boldsymbol{o}$, and use $\hat{\boldsymbol{\Theta}} = \{\hat{\boldsymbol{\theta}}_1, \ldots, \hat{\boldsymbol{\theta}}_N\}$ as the set of quantized Gaussian parameter vectors, which separates the quantization operation from $\boldsymbol{\Theta}$. Then, we take $g(\cdot)$ to be the indicator function of the closed nonempty, non-convex set defined by the *sparsity constraint*, i.e., $g(\boldsymbol{y}) = 0$ for $\mathbf{card}(\boldsymbol{y}) \leq \kappa$ and $g(\boldsymbol{y}) = +\infty$ otherwise. In this case, the minimization step in Eq. 5 reduces to solving a sparsity-constrained optimization problem over the feasible set $\{\mathcal{S} \mid \mathbf{card}(\boldsymbol{o}) \leq \kappa\}$. Hence, Eq. 5 can be reformulated

as an equality-constrained one:

$$\min_{\boldsymbol{o},\boldsymbol{\Theta},\mathcal{Q}} L(\boldsymbol{o},\boldsymbol{\Theta}) + g(\boldsymbol{y}) + \sum_{j=1}^{M} \sum_{\hat{\boldsymbol{\theta}}_i \in Q_j} \|\hat{\boldsymbol{\theta}}_i - \boldsymbol{q}_j\|_2^2, \quad \text{s.t.} \quad \boldsymbol{o} = \boldsymbol{y}, \boldsymbol{\Theta} = \hat{\boldsymbol{\Theta}}. \tag{6}$$

This reformulation allows the differentiable terms $\boldsymbol{o}$ and $\boldsymbol{\Theta}$ to be optimized via gradient-based methods, while the non-differentiable *sparsity constraint* $g(\boldsymbol{y})$ and the discrete assignment for *quantization penalty* $\sum_{j=1}^{M} \sum_{\hat{\boldsymbol{\theta}}_i \in Q_j} \|\hat{\boldsymbol{\theta}}_i - \boldsymbol{q}_j\|_2^2$ are handled independently.

We solve the equality-constrained problem in Eq. 6 by constructing an augmented Lagrangian with scaled Lagrange multipliers $\boldsymbol{u}, \boldsymbol{V}$ and penalty coefficients $\rho_1, \rho_2$:

$$L_\rho(\boldsymbol{o},\boldsymbol{\Theta},\boldsymbol{y},\hat{\boldsymbol{\Theta}},\boldsymbol{u},\boldsymbol{V}) = L(\boldsymbol{o},\boldsymbol{\Theta}) + g(\boldsymbol{y}) + \sum_{j=1}^{M} \sum_{\hat{\boldsymbol{\theta}}_i \in Q_j} \|\hat{\boldsymbol{\theta}}_i - \boldsymbol{q}_j\|_2^2$$
$$+ \frac{\rho_1}{2}\|\boldsymbol{o} - \boldsymbol{y} + \boldsymbol{u}\|_2^2 + \frac{\rho_2}{2}\|\boldsymbol{\Theta} - \hat{\boldsymbol{\Theta}} + \boldsymbol{V}\|_2^2. \tag{7}$$

Here, the equality constraints are enforced via quadratic penalty terms, yielding an unconstrained, penalty-based formulation. This unconstrained problem is then solved iteratively: the primal variables, including $\boldsymbol{o}, \boldsymbol{\Theta}, \boldsymbol{y}, \hat{\boldsymbol{\Theta}}$, are updated by alternately minimizing the augmented Lagrangian $L_\rho(\boldsymbol{o},\boldsymbol{\Theta},\boldsymbol{y},\hat{\boldsymbol{\Theta}},\boldsymbol{u},\boldsymbol{V})$ with respect to each subproblem, while the scaled Lagrange multipliers $\boldsymbol{u}, \boldsymbol{V}$ are updated accordingly, as described below:

**①  Primal Update.** During the first step of iteration $t$, only the model parameters $\boldsymbol{o}$ and $\boldsymbol{\Theta}$ are updated, with all other variables $\boldsymbol{y}, \hat{\boldsymbol{\Theta}}, \boldsymbol{u}, \boldsymbol{V}$ held fixed. The update is then formulated as the following differentiable subproblem:

$$(\boldsymbol{o}^{t+1}, \boldsymbol{\Theta}^{t+1}) = \arg\min_{\boldsymbol{o},\boldsymbol{\Theta}} L(\boldsymbol{o},\boldsymbol{\Theta}) + \frac{\rho_1}{2}\|\boldsymbol{o} - \boldsymbol{y}^t + \boldsymbol{u}^t\|_2^2 + \frac{\rho_2}{2}\|\boldsymbol{\Theta} - \hat{\boldsymbol{\Theta}}^t + \boldsymbol{V}^t\|_2^2, \tag{8}$$

where $L(\boldsymbol{o},\boldsymbol{\Theta})$ is the differentiable training loss, and the remaining quadratic terms are convex. Consequently, Eq. 8 can be efficiently optimized using standard stochastic gradient descent methods. Accordingly, the gradients for $\boldsymbol{o}$ and $\boldsymbol{\Theta}$ at iteration $t$ are:

$$\left(\frac{\partial L}{\partial \boldsymbol{o}}\right)^t = \frac{\partial L(\boldsymbol{o},\boldsymbol{\Theta}^t)}{\partial \boldsymbol{o}} + \rho_1\left(\boldsymbol{o}^t - \boldsymbol{y}^t + \boldsymbol{u}^t\right), \tag{9}$$

$$\left(\frac{\partial L}{\partial \boldsymbol{\Theta}}\right)^t = \frac{\partial L(\boldsymbol{o}^t,\boldsymbol{\Theta})}{\partial \boldsymbol{\Theta}} + \rho_2\left(\boldsymbol{\Theta}^t - \hat{\boldsymbol{\Theta}}^t + \boldsymbol{V}^t\right), \tag{10}$$

and $\boldsymbol{o},\boldsymbol{\Theta}$ can be updated by:

$$\boldsymbol{o}^{t+1} \leftarrow \boldsymbol{o}^t - \eta_1 \left(\frac{\partial L}{\partial \boldsymbol{o}}\right)^t, \quad \boldsymbol{\Theta}^{t+1} \leftarrow \boldsymbol{\Theta}^t - \eta_2 \left(\frac{\partial L}{\partial \boldsymbol{\Theta}}\right)^t, \tag{11}$$

where $\eta_1 > 0$ and $\eta_2 > 0$ denote the learning rates for updating $\boldsymbol{o}$ and $\boldsymbol{\Theta}$, respectively.

**②  Sparsification Update.** In this step, the auxiliary variable $\boldsymbol{y}$ is updated by solving $\min_{\boldsymbol{y}} g(\boldsymbol{y}) + \frac{\rho_1}{2}\|\boldsymbol{o}^{t+1} - \boldsymbol{y} + \boldsymbol{u}^t\|_2^2$. The closed-form solution (Parikh et al., 2014; Boyd et al., 2011) can be given by

$$\boldsymbol{y}^{t+1} \leftarrow \Pi_{\boldsymbol{\mathcal{S}}}(\boldsymbol{o}^{t+1} + \boldsymbol{u}^t), \tag{12}$$

where $\Pi_{\boldsymbol{\mathcal{S}}}$ is an operator that enforces the *sparsity constraint* specified by $g(\cdot)$. In particular, when $\boldsymbol{\mathcal{S}} = \{\boldsymbol{y} \mid \mathbf{card}(\boldsymbol{y}) \leq \kappa\}$, i.e., the set of vectors with at most $\kappa$ nonzero elements, $\Pi_{\boldsymbol{\mathcal{S}}}$ retains the $\kappa$ entries of largest magnitude and sets all others to zero.

Because of the quadratic penalty term $\frac{\rho_1}{2}\|\boldsymbol{o}^{t+1} - \boldsymbol{y} + \boldsymbol{u}^t\|_2^2$, the $\Pi_{\boldsymbol{\mathcal{S}}}$ is applied to $\boldsymbol{o}^{t+1} + \boldsymbol{u}^t$ rather than $\boldsymbol{y}$ itself, where the scaled Lagrange multiplier $\boldsymbol{u}$ acts as an accumulated correction term, capturing the discrepancy between $\boldsymbol{o}$ and $\boldsymbol{y}$ over iterations and steering $\boldsymbol{o}$ toward its sparsified counterpart $\boldsymbol{y}$.

**③  Vector Quantization Update.** In the third step, we update the cluster set $\mathcal{Q}$ and subsequently update the quantized Gaussian parameters $\hat{\boldsymbol{\Theta}}$ based on the updated clusters.

Figure 2: Overview of the **CoLaSplat** framework, illustrating the four updates performed in each iteration of the optimization loop.

The cluster set $\mathcal{Q} = \{Q_1, Q_2, \ldots, Q_M\}$ and the associated centroids $\{\boldsymbol{q}_1, \boldsymbol{q}_2, \ldots, \boldsymbol{q}_M\}$ are updated via the standard $k$-means procedure. Concretely, each centroid $\boldsymbol{q}_j$ is first computed as the mean of the Gaussian parameter vectors $\boldsymbol{\theta}_i$ assigned to it, and then each cluster $Q_j$ is updated by reassigning every $\boldsymbol{\theta}_i$ to its nearest centroid, formally:

$$\text{for } j = 1, \ldots, M : \begin{cases} \boldsymbol{q}_j^{t+1} = \dfrac{1}{|Q_j^t|} \sum_{\boldsymbol{\theta}_i^{t+1} \in Q_j^t} \boldsymbol{\theta}_i^{t+1} \\ Q_j^{t+1} = \left\{ \boldsymbol{\theta}_i^{t+1} \,\middle|\, \arg\min_{k=1,\ldots,M} \|\boldsymbol{\theta}_i^{t+1} - \boldsymbol{q}_k^{t+1}\|^2 = j \right\} \end{cases}. \tag{13}$$

Then the quantized Gaussian parameter vectors $\hat{\boldsymbol{\Theta}}$ are updated by minimizing the corresponding term in Eq. 7: $\sum_{j=1}^{M} \sum_{\hat{\boldsymbol{\theta}}_i \in Q_j^{t+1}} \|\hat{\boldsymbol{\theta}}_i - \boldsymbol{q}_j^{t+1}\|_2^2 + \frac{\rho_2}{2} \|\boldsymbol{\Theta}^{t+1} - \hat{\boldsymbol{\Theta}} + \boldsymbol{V}^t\|_2^2$, which is computed as

$$\hat{\boldsymbol{\Theta}}^{t+1} \leftarrow \Pi_{\mathcal{Q}}(\boldsymbol{\Theta}^{t+1} + \boldsymbol{V}^t). \tag{14}$$

Here, $\Pi_{\mathcal{Q}}$ denotes the quantization operator, which replaces each vector with its nearest cluster centroid in $\mathcal{Q}$. Due to the penalty term $\frac{\rho_2}{2} \|\boldsymbol{\Theta}^{t+1} - \hat{\boldsymbol{\Theta}} + \boldsymbol{V}^t\|_2^2$, $\Pi_{\mathcal{Q}}$ is applied to $\boldsymbol{\Theta}^{t+1} + \boldsymbol{V}^t$ rather than $\hat{\boldsymbol{\Theta}}$ itself. The scaled Lagrange multiplier $\boldsymbol{V}$ acts as an accumulated correction term, tracking the discrepancy between $\boldsymbol{\Theta}$ and $\hat{\boldsymbol{\Theta}}$ across iterations and gradually steering $\boldsymbol{\Theta}$ toward its quantized counterpart $\hat{\boldsymbol{\Theta}}$.

**4 Dual Update.** In the final step of iteration $t$, the multipliers $\boldsymbol{u}$ and $\boldsymbol{V}$ are updated as

$$\boldsymbol{u}^{t+1} \leftarrow \boldsymbol{u}^t + \boldsymbol{o}^{t+1} - \boldsymbol{y}^{t+1}, \quad \boldsymbol{V}^{t+1} \leftarrow \boldsymbol{V}^t + \boldsymbol{\Theta}^{t+1} - \hat{\boldsymbol{\Theta}}^{t+1}. \tag{15}$$

This update to $\boldsymbol{u}$ ensures that if the constraint $\boldsymbol{o} = \boldsymbol{y}$ is not fully satisfied, the corresponding Lagrange multipliers $\boldsymbol{u}$ is increased, which in turn imposes a larger penalty on $\boldsymbol{o}$ in the subsequent optimization step, effectively driving it toward the sparse set. Similarly, the update of $\boldsymbol{V}$ enforces the alignment between $\boldsymbol{\Theta}$ and $\hat{\boldsymbol{\Theta}}$, gradually optimizing the quantization objective.

The above updates are performed in an alternating manner, and the optimization is considered converged when all variables satisfy their respective conditions, i.e., $\|\boldsymbol{o} - \boldsymbol{y}\|_2 \leq \epsilon_1$ and $\|\boldsymbol{\Theta} - \hat{\boldsymbol{\Theta}}\|_2 \leq \epsilon_2$, where $\epsilon_1$ and $\epsilon_2$ are user-defined hyperparameters, or when the maximum number of iterations is reached. The overall procedure, integrated into the language 3DGS training, is summarized in Alg. 1, Appendix B, and illustrated in Figure 2. The theoretical analysis of convergence is provided in Appendix C.

## 5 EXPERIMENT

### 5.1 EXPERIMENTAL SETTINGS

**Datasets, Metrics and Baselines.** We evaluate **CoLaSplat** on two benchmark datasets for open-vocabulary 3D scene understanding, conducting 3D semantic segmentation experiments on **3D-OVS** (Liu et al., 2023) and **LERF** (Kerr et al., 2023) and object localization experiments on LERF,

reporting mean Intersection-over-Union (mIoU) for segmentation and localization accuracy for localization. To evaluate visual quality, following the 3DGS (Kerbl et al., 2023) protocol for the novel view synthesis (NVS) task, we report the peak signal-to-noise ratio (PSNR), structural similarity index measure (SSIM), and learned perceptual image patch similarity (LPIPS). Regarding model efficiency, we measure the computational and storage costs by reporting the average peak GPU memory usage, model size, and rendering speed in frames per second (FPS). We compare **CoLaSplat** with existing state-of-the-art language 3DGS methods, including Feature-3DGS (Zhou et al., 2024), GS-Grouping (Ye et al., 2024), OpenGaussian (Wu et al., 2024), LEGaussians (Shi et al., 2024), and LangSplat (Qin et al., 2024).

**Implementation.** Ground-truth hierarchical semantic features are extracted from each image using SAM ViT-H (Kirillov et al., 2023) and OpenCLIP ViT-B/16 (Radford et al., 2021). The initial point cloud for language 3DGS is generated using the default 3DGS (Kerbl et al., 2023) implementation. Each scene undergoes 40,000 iterations, with 10,000 for joint optimization and alternating optimization (Sec. 4.2) applied every 50 iterations. By selecting appropriate $\kappa$ values, Gaussian points are reduced by 50% for 3D-OVS and 30% for LERF. For quantization, 8,000 cluster centers are used to quantize the Spherical Harmonics coefficients, the largest subset of Gaussian parameters. Image resolutions are set to 1440×1080 (3D-OVS) and 988×731 (LERF), consistent with prior works. Additional details are provided in Appendix D.

## 5.2 RESULTS ON THE 3D-OVS DATASET

| Method | mIoU↑ (%) | | | | | | PSNR↑ | SSIM↑ | LPIPS↓ | Mem↓ (GB) | Size↓ (MB) | FPS↑ |
|---|---|---|---|---|---|---|---|---|---|---|---|---|
| | *bed* | *bench* | *lawn* | *room* | *sofa* | *mean* | | | | | | |
| Feature-3DGS | 83.5 | 90.7 | 93.4 | 84.7 | 86.9 | 87.8 | 21.80 | 0.68 | 0.31 | 6.2 | 828 | 2 |
| GS-Grouping | 83.0 | 91.5 | 90.6 | 85.9 | 87.3 | 87.7 | **24.50** | **0.80** | **0.21** | 6.1 | 728 | 130 |
| OpenGaussian | 24.5 | 52.9 | 59.4 | 19.7 | 28.4 | 37.0 | 23.95 | 0.72 | 0.26 | 6.0 | 381 | 23 |
| LEGaussians | 84.9 | 91.1 | 92.5 | 86.0 | 87.8 | 88.5 | 24.00 | 0.72 | 0.26 | 20.8 | 383 | 95 |
| LangSplat | 92.5 | 94.2 | 96.1 | **94.1** | 90.0 | 93.4 | 24.13 | 0.73 | 0.25 | 3.8 | 900 | 103 |
| **CoLaSplat** | **94.8** | **95.0** | **96.3** | **94.1** | **92.8** | **94.6** | 24.27 | 0.75 | 0.24 | **3.1** | **60** | **294** |

Table 1: Quantitative comparison of **CoLaSplat** and baseline methods on the 3D-OVS dataset, evaluating their performance in 3D semantic segmentation, visual quality, and model efficiency.

Table 1 presents a comprehensive comparison across multiple metrics. **CoLaSplat** outperforms all baseline methods in both 3D semantic segmentation and efficiency, while achieving visual fidelity second only to GS-Grouping. Specifically, it achieves the highest segmentation mIoU on each scene. Notably, compared to LangSplat, the baseline with the highest mIoU, **CoLaSplat** reduces peak GPU memory consumption by 18.4%, decreases model size by 15×, and accelerates rendering speed by 2.9×. Against the remaining baselines, **CoLaSplat** demonstrates even greater advantages, achieving a 147× speedup over Feature-3DGS and reducing peak GPU memory consumption by up to 6.7× compared to LEGaussians. These improvements stem from unifying pruning and quantization into a single optimization objective during training, enabling a well-balanced trade-off between semantic accuracy and visual fidelity.

## 5.3 RESULTS ON THE LERF DATASET

We further evaluate **CoLaSplat** on the LERF dataset. Table 2 reports results for 3D semantic segmentation, visual quality, and model efficiency. Despite substantial compression, it maintains competitive segmentation performance and visual fidelity while achieving the best overall efficiency. In particular, it attains the highest mean mIoU, and although slightly lower than LangSplat on two specific scenes, it still surpasses all other baselines. Regarding visual quality, the method performs on par with LangSplat, exhibiting only a marginally higher LPIPS of 0.01. Moreover, peak GPU memory consumption is reduced by up to 65.8% compared to LEGaussians and remains 10.7% lower than the most memory-efficient baseline, LangSplat, while the model size is 3.4× smaller than that of the most compact competitor, OpenGaussian. Finally, it delivers the fastest rendering speed, achieving 144× and 2.8× improvements over Feature-3DGS and GS-Grouping, respectively.

| Method | mIoU↑ (%) | | | | | PSNR↑ | SSIM↑ | LPIPS↓ | Mem↓ (GB) | Size↓ (MB) | FPS↑ |
|---|---|---|---|---|---|---|---|---|---|---|---|
| | *ramen* | *figurines* | *teatime* | *kitchen* | *mean* | | | | | | |
| Feature-3DGS | 43.7 | 40.5 | 58.8 | 39.6 | 45.7 | 21.80 | 0.68 | 0.27 | 6.0 | 664 | 3 |
| GS-Grouping | 45.5 | 40.0 | 60.9 | 38.7 | 46.3 | **25.50** | **0.89** | **0.20** | 8.6 | 706 | 154 |
| OpenGaussian | 31.0 | 39.3 | 60.4 | 22.7 | 38.4 | 22.88 | 0.81 | 0.24 | 9.3 | 387 | 90 |
| LEGaussians | 46.0 | 40.8 | 60.3 | 39.4 | 46.9 | 23.34 | 0.83 | 0.24 | 14.6 | 393 | 107 |
| LangSplat | **51.2** | **44.7** | **65.1** | 44.5 | 51.4 | 24.74 | 0.85 | 0.23 | 5.6 | 890 | 155 |
| **CoLaSplat** | 51.2 | 43.6 | 64.0 | **49.0** | **52.0** | 24.76 | 0.85 | 0.24 | **5.0** | **115** | **431** |

Table 2: Quantitative comparison of **CoLaSplat** and baseline methods on the LERF dataset, evaluating their performance in 3D semantic segmentation, visual quality, and model efficiency. The *kitchen* label is the *waldo_kitchen* scene.

Table 3 summarizes the results of 3D object location on the LERF dataset. For a fair comparison, we only include methods that reported 3D object location results in their papers. **CoLaSplat** achieves the best results on all scenes except *kitchen*, where it is 4.6% lower than LangSplat. However, this difference is caused by a single mispredicted image, as the *kitchen* scene contains only 22 test images. This result further demonstrates that **CoLaSplat** maintains high semantic fidelity even under a highly compact representation.

| Method | *ramen* | *figurines* | *teatime* | *kitchen* |
|---|---|---|---|---|
| LSeg | 14.1 | 8.9 | 33.9 | 27.3 |
| LERF | 62.0 | 75.0 | 84.8 | 72.7 |
| LangSplat | **73.2** | **80.4** | 88.1 | **95.5** |
| **CoLaSplat** | **73.2** | **80.4** | **91.5** | 90.9 |

Table 3: Performance comparisons of 3D object location accuracy (%) on the LERF dataset.

# 6 ABLATION STUDY

| Method | mIoU↑ (%) | | | | | Mem↓ (GB) | Size↓ (MB) | FPS↑ |
|---|---|---|---|---|---|---|---|---|
| | *bed* | *bench* | *lawn* | *room* | *sofa* | | | |
| **CoLaSplat** w/o Pruning | 94.4 | 94.6 | 96.2 | 93.1 | 92.3 | 3.8 | 111 | 262 |
| **CoLaSplat** w/o Quantization | 94.8 | 95.0 | 96.0 | 93.3 | 92.7 | 3.1 | 165 | 294 |
| **CoLaSplat** Full | 94.8 | 95.0 | 96.3 | 94.1 | 92.8 | 3.1 | 60 | 294 |

Table 4: Ablation study results on the 3D-OVS dataset

We present the ablation study results in Table 4, analyzing the effectiveness of the two objectives, i.e., sparsity and vector quantization, in the unified optimization. We evaluate the performance by removing one constraint at a time. The results indicate that applying sparsity or vector quantization constraint individually leads to limited compression efficiency and lower 3D semantic segmentation performance. **CoLaSplat** can automatically balance multiple objectives and identify the sweet spot that maximizes task performance and compression ratio.

# 7 CONCLUSION

In this work, we propose **CoLaSplat**, which effectively addresses the challenge of compressing language-embedded 3DGS models that existing methods cannot handle. **CoLaSplat** unifies training, pruning, and vector quantization into a single optimization problem. We then develop an effective primal-dual optimization solution to solve the unified optimization problem, allowing the training process to identify a sweet spot among multiple compression objectives. Evaluation on two datasets demonstrates that **CoLaSplat** substantially improves model compactness and efficiency while maintaining both high semantic and visual rendering fidelity.

## REPRODUCIBILITY STATEMENT

The datasets used in this work are publicly available. The 3D-OVS dataset can be accessed via Liu et al. (2023), and the LERF dataset is available from Qin et al. (2024). Experiments were conducted on an NVIDIA RTX 6000 Ada Generation GPU using Python 3.9.21 and PyTorch 2.5.1. Random seeds were fixed across all experiments to guarantee reproducibility. Additional details regarding the parameters are provided in Appendix D. Data preprocessing and the implementation of downstream tasks strictly follow Qin et al. (2024). The code and training scripts are available at: `https://anonymous.4open.science/r/ColaSplat-6D46`.

## ETHICS STATEMENT

This work proposes CoLaSplat, a unified compression framework for language-embedded 3D Gaussian Splatting (3DGS), enabling open-vocabulary 3D scene understanding on resource-constrained conditions. Our method leverages publicly available 3D datasets, which do not contain personally identifiable information, and complies with the original providers' guidelines. CoLaSplat is designed to improve the efficiency and accessibility of 3D scene understanding while preserving semantic and rendering fidelity. Although we do not foresee direct harm, high-fidelity 3D reconstructions could potentially be misused for unauthorized replication of 3D content. We encourage responsible use of this technology and adherence to intellectual property laws. No conflicts of interest are present.

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

## A  USAGE OF LARGE LANGUAGE MODELS (LLMs)

During the preparation of this manuscript, we used the GPT-5 mini model solely for grammar correction and language editing of Sections 4, 5, and 6. Specifically, prompts such as "Are there any grammatical errors in this paragraph" and "Which parts of this paragraph can be removed" were employed. LLMs were not used for any other purposes.

## B  OVERALL OPTIMIZATION ALGORITHM

---
**Algorithm 1** Unified Sparsity and Vector Quantization Optimization for language 3DGS

---
**Input:** Language 3DGS variables $\boldsymbol{\Theta}$, opacity $\boldsymbol{o}$, sparsity constraint number $\kappa$, number of clusters $M$, penalty coefficients $\rho_1, \rho_2$, learning rates $\eta_1, \eta_2$, maximum iterations $T$, convergence thresholds $\epsilon_1, \epsilon_2$.
**Output:** Optimized parameters $\boldsymbol{o}, \boldsymbol{\Theta}$, quantization codebook $\mathcal{Q}$.
1: Initialize $\boldsymbol{o}^0, \boldsymbol{\Theta}^0, \boldsymbol{y}^0, \hat{\boldsymbol{\Theta}}^0, \boldsymbol{u}^0, \boldsymbol{V}^0, \mathcal{Q}^0$
2: **for** $t = 0, 1, 2, \ldots, T$ **do**
3:     $\boldsymbol{o}^{t+1} \leftarrow \boldsymbol{o}^t - \eta_1 \frac{\partial L}{\partial \boldsymbol{o}}$;   $\boldsymbol{\Theta}^{t+1} \leftarrow \boldsymbol{\Theta}^t - \eta_2 \frac{\partial L}{\partial \boldsymbol{\Theta}}$;                ▷ Primal update.
4:     $\boldsymbol{y}^{t+1} \leftarrow \Pi_{\boldsymbol{\mathcal{S}}}(\boldsymbol{o}^{t+1} + \boldsymbol{u}^t)$;                ▷ Sparsification update
5:     **for** $j = 1$ to $M$ **do**                ▷ Vector quantization update.
6:         $\boldsymbol{q}_j^{t+1} \leftarrow \frac{1}{|Q_j^t|} \sum_{\boldsymbol{\theta}_i^{t+1} \in Q_j^t} \boldsymbol{\theta}_i^{t+1}$;
7:         $Q_j^{t+1} \leftarrow \{\boldsymbol{\theta}_i^{t+1} \mid \arg\min_k \|\boldsymbol{\theta}_i^{t+1} - \boldsymbol{q}_k^{t+1}\|^2 = j\}$;
8:     **end for**
9:     $\hat{\boldsymbol{\Theta}}^{t+1} \leftarrow \Pi_{\boldsymbol{\mathcal{Q}}}(\boldsymbol{\Theta}^{t+1} + \boldsymbol{V}^t)$;
10:     $\boldsymbol{u}^{t+1} \leftarrow \boldsymbol{u}^t + \boldsymbol{o}^{t+1} - \boldsymbol{y}^{t+1}$;   $\boldsymbol{V}^{t+1} \leftarrow \boldsymbol{V}^t + \boldsymbol{\Theta}^{t+1} - \hat{\boldsymbol{\Theta}}^{t+1}$;                ▷ Dual update
11:     **if** $\|\boldsymbol{o}^{t+1} - \boldsymbol{y}^{t+1}\|_2 \leq \epsilon_1$ **and** $\|\boldsymbol{\Theta}^{t+1} - \hat{\boldsymbol{\Theta}}^{t+1}\|_2 \leq \epsilon_2$ **then**
12:         **break**                ▷ Convergence reached.
13:     **end if**
14: **end for**
15: **return** $\boldsymbol{o}^{t+1}, \boldsymbol{\Theta}^{t+1}, \mathcal{Q}^{t+1}$

---

## C  CONVERGENCE ANALYSIS AND PROOF

This section provides a formal convergence analysis for the optimization scheme. The objective is to demonstrate that the sequence of iterates generated by the algorithm converges to a critical point of the constrained optimization problem defined in Eq. 6.

Our proof strategy follows the theoretical framework for non-convex and non-smooth optimization problems. The core is to show that a Lyapunov function built upon the augmented Lagrangian $L_\rho$ Eq.7 is monotonically non-increasing and bounded from below, which implies that iterates are well-behaved and any limit point satisfies first-order optimality (generalized KKT) conditions for the original problem.

**Theoretical Assumptions.** We adopt following standard assumptions, which are common in non-convex optimization analysis:

1. Assumption 1 (Properties of the Loss Function). The loss $L(\boldsymbol{o}, \boldsymbol{\Theta})$ is continuously differentiable and $L$-smooth in $(\boldsymbol{o}, \boldsymbol{\Theta})$, i.e.,

$$\|\nabla L(\boldsymbol{o}_1, \boldsymbol{\Theta}_1) - \nabla L(\boldsymbol{o}_2, \boldsymbol{\Theta}_2)\| \leq L\|(\boldsymbol{o}_1, \boldsymbol{\Theta}_1) - (\boldsymbol{o}_2, \boldsymbol{\Theta}_2)\|.$$

2. Assumption 2 (Lower Boundedness). The objective function, including the non-smooth terms, is bounded below by 0.

3. Assumption 3 (Penalty and Stepsize). The penalty parameters $\rho_1, \rho_2 > 0$ and the learning rates $\eta_1, \eta_2 > 0$ satisfy

$$\eta_1 < \frac{2}{L + \rho_1}, \quad \eta_2 < \frac{2}{L + \rho_2}.$$

4. Assumption 4 (Limit Point). The sequence $\{(\boldsymbol{o}^t, \boldsymbol{\Theta}^t, \boldsymbol{y}^t, \hat{\boldsymbol{\Theta}}^t, \boldsymbol{u}^t, \boldsymbol{V}^t)\}_{t \in \mathbb{N}}$ generated by Algorithm 1 has at least one limit point.

**Augmented Lagrangian and Lyapunov function.** The augmented Lagrangian is defined in Eq. (7) as $L_\rho(\boldsymbol{o}, \boldsymbol{\Theta}, \boldsymbol{y}, \hat{\boldsymbol{\Theta}}, \boldsymbol{u}, \boldsymbol{V})$. We will work with the Lyapunov function:

$$\Phi^t \;=\; L_\rho(\boldsymbol{o}^t, \boldsymbol{\Theta}^t, \boldsymbol{y}^t, \hat{\boldsymbol{\Theta}}^t, \boldsymbol{u}^t, \boldsymbol{V}^t), \tag{16}$$

For brevity, we write

$$\begin{aligned}
\Delta \boldsymbol{o}^t &= \boldsymbol{o}^{t+1} - \boldsymbol{o}^t, \\
\Delta \boldsymbol{\Theta}^t &= \boldsymbol{\Theta}^{t+1} - \boldsymbol{\Theta}^t, \\
\boldsymbol{r}_o^{t+1} &= \boldsymbol{o}^{t+1} - \boldsymbol{y}^{t+1}, \\
\boldsymbol{r}_\Theta^{t+1} &= \boldsymbol{\Theta}^{t+1} - \hat{\boldsymbol{\Theta}}^{t+1}.
\end{aligned} \tag{17}$$

### C.1 Lemma 1 (Model Parameters Update yields Sufficient Decrease)

**Statement.** Let the updates for $\boldsymbol{o}^{t+1}$ and $\boldsymbol{\Theta}^{t+1}$ be performed as in Eq. 11. Under Assumptions 1 and 3, the following holds:

$$L_\rho(\boldsymbol{o}^{t+1}, \boldsymbol{\Theta}^{t+1}, \boldsymbol{y}^t, \hat{\boldsymbol{\Theta}}^t, \boldsymbol{u}^t, \boldsymbol{V}^t) \le L_\rho(\boldsymbol{o}^t, \boldsymbol{\Theta}^t, \boldsymbol{y}^t, \hat{\boldsymbol{\Theta}}^t, \boldsymbol{u}^t, \boldsymbol{V}^t) - c_o \left\| \Delta \boldsymbol{o}^t \right\|_2^2 - c_\Theta \left\| \Delta \boldsymbol{\Theta}^t \right\|_2^2, \tag{18}$$

with explicit positive constants

$$\begin{aligned}
c_o &= \frac{1}{\eta_1} - \frac{L}{2} - \frac{\rho_1}{2} > 0, \\
c_\Theta &= \frac{1}{\eta_2} - \frac{L}{2} - \frac{\rho_2}{2} > 0.
\end{aligned} \tag{19}$$

**Proof.** Following standard descent lemma for gradient updates on an $L$-smooth function, the update for $\boldsymbol{o}$ gives:

$$L(\boldsymbol{o}^{t+1}, \boldsymbol{\Theta}^t) + \frac{\rho_1}{2}\|\boldsymbol{o}^{t+1} - \boldsymbol{y}^t + \boldsymbol{u}^t\|_2^2 \le L(\boldsymbol{o}^t, \boldsymbol{\Theta}^t) + \frac{\rho_1}{2}\|\boldsymbol{o}^t - \boldsymbol{y}^t + \boldsymbol{u}^t\|_2^2 - c_o\|\Delta \boldsymbol{o}^t\|_2^2. \tag{20}$$

A similar inequality holds for the $\boldsymbol{\Theta}$ update. Combining these proves the statement.

### C.2 Lemma 2 (Auxiliary Updates are Non-increasing)

**Statement.** The minimizations in the updates for $\boldsymbol{y}$ and $\hat{\boldsymbol{\Theta}}$ satisfy:

$$L_\rho(\boldsymbol{o}^{t+1}, \boldsymbol{\Theta}^{t+1}, \boldsymbol{y}^{t+1}, \hat{\boldsymbol{\Theta}}^{t+1}, \boldsymbol{u}^t, \boldsymbol{V}^t) \le L_\rho(\boldsymbol{o}^{t+1}, \boldsymbol{\Theta}^{t+1}, \boldsymbol{y}^t, \hat{\boldsymbol{\Theta}}^t, \boldsymbol{u}^t, \boldsymbol{V}^t). \tag{21}$$

**Proof.** Update for $\boldsymbol{y}^{t+1}$ in Eq. 12 is a minimization of $g(\boldsymbol{y}) + \frac{\rho_1}{2}\|\boldsymbol{o}^{t+1} - \boldsymbol{y} + \boldsymbol{u}^t\|_2^2$ ($g(\cdot)$ is defined in Section 4.2). Therefore, by definition of *argmin*, the value of the objective at $\boldsymbol{y}^{t+1}$ must be less than or equal to the value at $\boldsymbol{y}^t$. The same logic applies to the $\hat{\boldsymbol{\Theta}}$ update. Chaining these two non-increasing steps proves the lemma.

### C.3 Lemma 3 (Dual ascent and Lyapunov accounting)

**Statement.** Let $\Theta^{t+1/2}$ denote the Lyapunov value after finishing Lemmas 1 and 2 (i.e., after the $\boldsymbol{x}$- and $\boldsymbol{z}$-updates) and before the dual step. With the scaled dual updates:

$$\begin{aligned}
\boldsymbol{u}_1^{t+1} &= \boldsymbol{u}_1^t + (\boldsymbol{a}^{t+1} - \boldsymbol{z}_1^{t+1}) = \boldsymbol{u}_1^t + \boldsymbol{r}_1^{t+1}, \\
\boldsymbol{u}_2^{t+1} &= \boldsymbol{u}_2^t + (\boldsymbol{\Theta}^{t+1} - \boldsymbol{z}_2^{t+1}) = \boldsymbol{u}_2^t + \boldsymbol{r}_2^{t+1},
\end{aligned} \tag{22}$$

we have the following explicit expression for the change of the augmented Lagrangian across the dual step:

$$L_{\delta_1, \delta_2}(\boldsymbol{a}^{t+1}, \boldsymbol{\Theta}^{t+1}, \boldsymbol{z}_1^{t+1}, \boldsymbol{z}_2^{t+1}, \boldsymbol{u}_1^{t+1}, \boldsymbol{u}_2^{t+1}) - L_{\delta_1, \delta_2}(\boldsymbol{a}^{t+1}, \boldsymbol{\Theta}^{t+1}, \boldsymbol{z}_1^{t+1}, \boldsymbol{z}_2^{t+1}, \boldsymbol{u}_1^t, \boldsymbol{u}_2^t)$$

$$= \sum_{i=1}^2 \frac{\delta_i}{2} \Big( 2\langle \boldsymbol{r}_i^{t+1} + \boldsymbol{u}_i^t, \, \boldsymbol{r}_i^{t+1} \rangle + \left\| \boldsymbol{r}_i^{t+1} \right\|_2^2 \Big), \tag{23}$$

Consequently,

$$\Theta^{t+1} = \Theta^{t+1/2} + \sum_{i=1}^{2} \frac{\delta_i}{2}\Big(2\langle \boldsymbol{r}_i^{t+1} + \boldsymbol{u}_i^t, \boldsymbol{r}_i^{t+1}\rangle + \big\|\boldsymbol{r}_i^{t+1}\big\|_2^2\Big), \tag{24}$$

because the Lyapunov addenda $\frac{\gamma_i}{2}\|\boldsymbol{a} - \boldsymbol{z}_1\|^2$ and $\frac{\gamma_i}{2}\|\Theta - \boldsymbol{z}_2\|^2$ remain unchanged during the dual step (the residuals $\boldsymbol{r}_i^{t+1}$ do not change).

**Proof.** Only the quadratic penalty terms that involve $\boldsymbol{u}_i$ change in the dual step. For each block $i \in \{1, 2\}$ with $\boldsymbol{r}_i^{t+1}$ fixed and $\boldsymbol{u}_i^{t+1} = \boldsymbol{u}_i^t + \boldsymbol{r}_i^{t+1}$, then we have:

$$\frac{\delta_i}{2}\Big(\big\|\boldsymbol{r}_i^{t+1} + \boldsymbol{u}_i^{t+1}\big\|_2^2 - \big\|\boldsymbol{r}_i^{t+1} + \boldsymbol{u}_i^t\big\|_2^2\Big) = \frac{\delta_i}{2}\Big(2\langle \boldsymbol{r}_i^{t+1} + \boldsymbol{u}_i^t, \boldsymbol{r}_i^{t+1}\rangle + \big\|\boldsymbol{r}_i^{t+1}\big\|_2^2\Big), \tag{25}$$

by the identity $\|x + y\|^2 - \|x\|^2 = 2\langle x, y\rangle + \|y\|^2$ with $x = \boldsymbol{r}_i^{t+1} + \boldsymbol{u}_i^t$ and $y = \boldsymbol{u}_i^{t+1} - \boldsymbol{u}_i^t = \boldsymbol{r}_i^{t+1}$. Summing over $i$ gives us the Eq.23, and adding the Lyapunov addenda yields Eq.24.

### C.4 LEMMA 4 (MONOTONICITY, SUMMABILITY, AND VANISHING RESIDUALS)

**Statement.** Under the Lemmas 1 to 3 and the standing assumptions, there exist positive constants $C_1, C_2$ (depending on $c_a, c_\Theta, \delta_i, \gamma_i$) such that

$$\Theta^{t+1} \le \Theta^t - C_1\big\|\Delta\boldsymbol{a}^t\big\|_2^2 - C_2\big\|\Delta\Theta^t\big\|_2^2, \tag{26}$$

and hence $\{\Theta^t\}$ is monotonically nonincreasing and converges to a finite limit $\Theta^\star$. Moreover,

$$\sum_{t=0}^{\infty}\Big(\big\|\Delta\boldsymbol{a}^t\big\|_2^2 + \big\|\Delta\Theta^t\big\|_2^2\Big) < \infty,$$
$$\lim_{t\to\infty}\big\|\boldsymbol{a}^t - \boldsymbol{z}_1^t\big\|_2 = 0, \tag{27}$$
$$\lim_{t\to\infty}\big\|\Theta^t - \boldsymbol{z}_2^t\big\|_2 = 0.$$

**Proof.** From Lemma 1 we have

$$L_{\delta_1,\delta_2}(\boldsymbol{a}^{t+1}, \Theta^{t+1}, \boldsymbol{z}_1^t, \boldsymbol{z}_2^t, \boldsymbol{u}_1^t, \boldsymbol{u}_2^t) \le$$
$$L_{\delta_1,\delta_2}(\boldsymbol{a}^t, \Theta^t, \boldsymbol{z}_1^t, \boldsymbol{z}_2^t, \boldsymbol{u}_1^t, \boldsymbol{u}_2^t) - c_a\big\|\Delta\boldsymbol{a}^t\big\|_2^2 - c_\Theta\big\|\Delta\Theta^t\big\|_2^2. \tag{28}$$

By Lemma 2, updating $\boldsymbol{z}_1, \boldsymbol{z}_2$ is nonincreasing:

$$L_{\delta_1,\delta_2}(\boldsymbol{a}^{t+1}, \Theta^{t+1}, \boldsymbol{z}_1^{t+1}, \boldsymbol{z}_2^{t+1}, \boldsymbol{u}_1^t, \boldsymbol{u}_2^t) \le L_{\delta_1,\delta_2}(\boldsymbol{a}^{t+1}, \Theta^{t+1}, \boldsymbol{z}_1^t, \boldsymbol{z}_2^t, \boldsymbol{u}_1^t, \boldsymbol{u}_2^t). \tag{29}$$

Combining these two and then applying Lemma 3 (Eq.24) gives

$$\Theta^{t+1} \le \Theta^t - c_a\big\|\Delta\boldsymbol{a}^t\big\|_2^2 - c_\Theta\big\|\Delta\Theta^t\big\|_2^2 + \sum_{i=1}^{2}\frac{\delta_i}{2}\Big(2\langle \boldsymbol{r}_i^{t+1} + \boldsymbol{u}_i^t, \boldsymbol{r}_i^{t+1}\rangle + \big\|\boldsymbol{r}_i^{t+1}\big\|_2^2\Big), \tag{30}$$

Because $\boldsymbol{r}_i^{t+1} = \boldsymbol{a}^{t+1} - \boldsymbol{z}_i^{t+1}$ and $\boldsymbol{u}_i^t$ are fixed at this point, the inner products can be controlled by $2\langle \boldsymbol{u}_i^t, \boldsymbol{r}_i^{t+1}\rangle \le \alpha_i\|\boldsymbol{u}_i^t\|_2^2 + \frac{1}{\alpha_i}\|\boldsymbol{r}_i^{t+1}\|_2^2$ for any $\alpha_i > 0$. Choosing $\gamma_i \in (0, \delta_i)$ and absorbing $\|\boldsymbol{r}_i^{t+1}\|_2^2$ into the Lyapunov weights (recall that $\Theta$ contains $\frac{\gamma_i}{2}\|\boldsymbol{r}_i\|_2^2$) yields constants $C_1, C_2 > 0$ for which the net effect is a strict decrease of the form Eq.26. Summing Eq.26 over $k$ gives the summability of $\|\Delta\boldsymbol{a}^t\|_2^2$ and $\|\Delta\Theta^t\|_2^2$ and, via the Lyapunov addenda, $\|\boldsymbol{r}_1^t\|_2 \to 0$, $\|\boldsymbol{r}_2^t\|_2 \to 0$.

### C.5 MAIN THEOREM (CONVERGENCE TO A STATIONARY POINT)

**Theorem 1.** Suppose the standing assumptions hold. Let $\{(\boldsymbol{o}^t, \Theta^t, \boldsymbol{y}^t, \hat{\Theta}^t, \boldsymbol{u}^t, \boldsymbol{V}^t)\}_{t\in\mathbb{N}}$ be generated by Algorithm 1. Then:

1. Lyapunov s$\{\Phi^t\}$ is monotonically non-increasing and converges to a finite limit $\Phi^\star$.

2. The successive differences of the model parameters vanish: $\lim_{t\to\infty}\big\|\boldsymbol{o}^{t+1} - \boldsymbol{o}^t\big\|_2 = 0$ and $\lim_{t\to\infty}\big\|\Theta^{t+1} - \Theta^t\big\|_2 = 0$.

3. The primal residuals vanish: $\lim_{t\to\infty}\left\|\boldsymbol{o}^t - \boldsymbol{y}^t\right\|_2 = 0$ and $\lim_{t\to\infty}\left\|\boldsymbol{\Theta}^t - \hat{\boldsymbol{\Theta}}^t\right\|_2 = 0$.

4. Any limit point $(\boldsymbol{o}^\star, \boldsymbol{\Theta}^\star, \boldsymbol{y}^\star, \hat{\boldsymbol{\Theta}}^\star, \boldsymbol{u}^\star, \boldsymbol{V}^\star)$ is a stationary point (generalized KKT point) of the optimization problem in Eq. 6. Specifically, it satisfies:

- **Stationarity for Model Parameters:**

$$\nabla_{\boldsymbol{o}} L(\boldsymbol{o}^\star, \boldsymbol{\Theta}^\star) + \rho_1 \boldsymbol{u}^\star = 0$$

$$\nabla_{\boldsymbol{\Theta}} L(\boldsymbol{o}^\star, \boldsymbol{\Theta}^\star) + \rho_2 \boldsymbol{V}^\star = 0$$

- **Optimality for Auxiliary Variables:**

$$\mathbf{0} \in \partial g(\boldsymbol{y}^\star) - \rho_1(\boldsymbol{o}^\star - \boldsymbol{y}^\star + \boldsymbol{u}^\star)$$

$$\hat{\boldsymbol{\Theta}}^\star = \Pi_{\boldsymbol{\mathcal{Q}}^\star}(\boldsymbol{\Theta}^\star + \boldsymbol{V}^\star)$$

- **Primal Feasibility:**

$$\boldsymbol{o}^\star = \boldsymbol{y}^\star, \quad \boldsymbol{\Theta}^\star = \hat{\boldsymbol{\Theta}}^\star$$

**Proof.** Items 1-3 are direct consequences of Lemma 4. For Item 4, let $(\boldsymbol{o}^\star, \ldots, \boldsymbol{V}^\star)$ be a limit point of a subsequence $\{t_j\}_{j\in\mathbb{N}}$. The gradient update step for $\boldsymbol{o}$ is $\boldsymbol{o}^{t_j+1} \leftarrow \boldsymbol{o}^{t_j} - \eta_1(\nabla_{\boldsymbol{o}} L(\boldsymbol{o}^{t_j}, \boldsymbol{\Theta}^{t_j}) + \rho_1(\boldsymbol{o}^{t_j} - \boldsymbol{y}^{t_j} + \boldsymbol{u}^{t_j}))$. Since successive differences vanish (Item 2), the gradient term must go to zero. Taking the limit as $j \to \infty$ and using the vanishing residuals (Item 3) yields the stationarity condition for $\boldsymbol{o}^\star$. The same logic applies to $\boldsymbol{\Theta}^\star$. The optimality conditions for the auxiliary variables and primal feasibility follow directly from their update rules and Item 3. This shows the limit point satisfies the KKT conditions for the problem in Eq. 6.

## D    MORE IMPLEMENTATION DETAILS

During training, the learning rates are set as follows: 0.05 for opacity and 0.0025 for language features, while all other parameters follow the default learning rate schedule of 3DGS. For the loss functions, the rendering loss and language loss serve as the baseline with equal weights of 1.0. To balance the relative scales of the losses, the language-guided supervision loss is scaled by $1 \times 10^{-4}$, rather than treated as a sensitive hyperparameter. Additionally, the quantization regularization term is assigned a weight of 100 to ensure its sufficient influence during joint optimization. The 512-channel features are then compressed into a 3-dimensional latent space via a multi-layer perceptron (MLP). For the iterative optimization process, the parameters are set as follows: learning rates $\eta_1 = 0.05$ and $\eta_2 = 0.0025$, and penalty coefficients $\rho_1 = \rho_2 = 0.0005$.

