# OpenReview forum: "CoLaSplat: Compact Language 3D Gaussian Splatting"
_ICLR.cc/2026/Conference — ICLR 2026 Conference Withdrawn Submission_

### Official Review · Reviewer_8x7N · 2025-10-25

**Soundness:** 3
**Presentation:** 2
**Contribution:** 2
**Rating:** 2
**Confidence:** 5

**Summary:**

The paper proposes CoLaSplat, a unified compression framework for language-embedded 3D Gaussian Splatting (3DGS). Instead of applying pruning and quantization as separate post-hoc stages, the method formulates semantic learning + sparsification + vector quantization as a single constrained optimization and solves it with a four-step primal–dual scheme (primal update, sparsification via cardinality projection, k-means–style vector quantization, dual update).

**Strengths:**

1. This paper is well written and easy to follow.
2. The method itself is pretty simple yet efficient.
3. This paper is the first paper working on the language Gaussian Splatting compression, and got the state of the art results.
4. The paper gives a pretty comprehensive mathematical formuilation of the optimization process. Convergence reasoning provided in Appendix C.
5. Thanks for providing a code and show me what is the main logic in the code. This really helps me to understand the paper.

**Weaknesses:**

1. Results are on LERF and 3D-OVS; both are useful but relatively small. Adding more diverse scenes (e.g., indoor/outdoor mixes, cluttered small-object scenes) would better stress pruning effects on fine geometry/rare semantics. Please refer to SceneSplat [1] and SceneSplat++ [2] for more language-embeded 3dgs data.

2. Baseline results on 3D-OVS is pretty outdated. FMGS [3], LUDVIS [4], OccamLGS [5], and SceneSplat [1] are having better results now. LangSplat also has a V2. Can you please include the recent baselines?

3. The whole idea is pretty similar in 3DGS compression series works. For example, the K-means codebook is introduced in CompGS [6],  SUNDAE is pruning [7] and there's a lot of other methods in this field as well. I won't list them all. I did notice that author mentioned these methods in literature. However, what if you use these methods directly on the language 3DGS? This work seems to work on LangSplat like idea, which is per-gaussian language feature. And these 3dgs compression methods, like k-means vector quantization, or any other quantization could be easily added to language features, the geometry-based pruning or opacity-based pruning can be also easily applied. But author has no discussion what will happen with simply applying these method.

4. I didn't totally understand what is card(o) here and the meaning of cardinality of opacities. And what is the obstacle to make card(o) differentiable? What is this sparsity constraint different from directly pruning gaussians with high opacity?

5. I am very happy to see that in this community that someone introduces dynamic system and optimal control into 3DGS. However, from the Sec 4.2, this method seems to be applicable to any kind of gaussian splatting instead of language gaussian splatting. I didn't see any language-specific design, as opacity is treated as an important term and all of the other terms including gaussian parameters and language features are treated same.

6. Training LangSplat from my experience is pretty hard. Can you also report training time?

7. Thanks for providing the code. However, I cannot find the provided checkpoint mentioned in README.md.

[1] SceneSplat: Gaussian Splatting-based Scene Understanding With Vision-Language Pretraining.
[2] SceneSplat++: A Large Dataset and Comprehensive Benchmark for Language Gaussian Splatting.
[3] FMGS: Foundation Model Embedded 3D Gaussian Splatting for Holistic 3D Scene Understanding
[4] LUDVIG: Learning-Free Uplifting of 2D Visual Features to Gaussian Splatting Scenes
[5] Occam's LGS: An Efficient Approach for Language Gaussian Splatting
[6] CompGS: Smaller and Faster Gaussian Splatting with Vector Quantization
[7] SUNDAE: Spectrally Pruned Gaussian Fields with Neural Compensation

**Questions:**

1. Can you please specify how each part of the optimization affecting the results? I can understand that each part is important in the ablation study. But for each part, I can find some examples in compressed 3dgs paper (mentioned in the weaknees part) that have some easy implementation. Can you add a comparison of what will happen if you directly adopt those techniques?
2. Can you also compare with other 3DGS compression methods? Like directly apply your method on 3DGS, or 3DGS with inverse rendering features?
3. Other questions are already in the weakness part.

---

### Official Review · Reviewer_CXiq · 2025-10-30

**Soundness:** 2
**Presentation:** 2
**Contribution:** 2
**Rating:** 4
**Confidence:** 2

**Summary:**

This paper introduces a method for a language-embedded 3D Gaussian Splatting. Author observed that prior work raising memory concern for 3DGS or semantic field but compression for 3DGS geometry/appearance and for language features has largely been developed separately. Authors argue that these independent techniques are not optimal for language-embedded 3DGS and motivate a unified compression approach tailored to the joint representation. The main idea is a compression framework that considers the full characteristics of language-embedded 3DGS. They propose an iterative primal–dual optimization scheme and provide mathematical justification for its advantages in the target setting. On standard benchmarks, the authors claim that the method achieved notable gains in memory footprint and rendering speed.

**Strengths:**

- Writing quality: Clear exposition.
- Mathematical development: The derivations are well structured and easy to follow.

**Weaknesses:**

**Baselines may be too weak**
- If the claim is that prior compressions are not suitable for language-embedded 3DGS, then the minimum baselines should include:

  - A 3DGS compression method + a separate language-feature compression method combined together.

  - or at least, A LangSplat-style setup (pretrained 3DGS + fitted language embeddings) compared against your unified compression while updating only the language embeddings on top of a **compressed 3DGS**.
- These would better test whether the unified approach offers intrinsic advantages over “separate compressions.”

**Experimental explanations needed**
- Task mismatch with OpenGaussian.

  - OpenGaussian focuses on 3D activations, so a direct comparison to your Section 2 setup seems odd; please clarify evaluation alignment.
- PSNR discrepancy across LangSplat vs. OpenGaussian.

  - As far as I understand both methods are just for attaching language embeddings to the 3DGS. In this case, 3DGS itself should remains the same why do PSNR values differ? Please explain the training/inference protocols that lead to this gap.

**Minor points**
- Consider a related-work discussion vs. Dr. Splat and ICCV CF^2, which also touch on efficiency/representation trade-offs.
- Line 391: You mention selecting an appropriate κ; how is this value determined in practice?
- Training details : It is stated that joint and alternating optimization are repeated every 50 steps—does this mean 40,000 base steps with an additional 10,000 steps every 50 iterations, or something else? Please clarify the schedule precisely.
- Sparsification vs. densification: If both run during training, won’t they conflict? How does interleaving them mid-training compare to applying sparsification post-training on a fixed model?

**Questions:**

Addressed within the Weaknesses and Minor points above.

---

### Official Review · Reviewer_M3pa · 2025-10-31

**Soundness:** 3
**Presentation:** 1
**Contribution:** 2
**Rating:** 2
**Confidence:** 4

**Summary:**

This paper proposes an algorithm for learning language embeddings on top of 3D Gaussian representation. While the previous studies often takes two training stages, one for optimizing 3D Gaussians and the other for embedding language features, the paper merges these steps into a single optimization step.

**Strengths:**

First of all, this paper produces a promising result in reduced model size, faster inference, and low memory usage. By optimizing 3D Gaussian by minimizing the RGB rendering loss as well as proposed losses, this paper remarkably prunes out the redundant Gaussians, which I believe the reason for the efficiency of this paper.

While the authors claim that the 'co-optimization of semantic learning, sparsification, and vector quantization' are novel, but I cannot clearly catch its novelty. I will describe my observation in the next section.

**Weaknesses:**

Overall, the authors put lots of efforts on description of the single optimization strategy. However, I cannot clearly catch the overall training pipeline (__Q3__) due to lots of missing details (__Q3__, __Q2__, __W3__). Moreover the experiment setup is problematic (__W1__, __W3__), and I do want to know about the theoretical necessity of the co-optimization beyond the experimental / empirical results (__W2__).

__W1. Unfair comparison__
The comparison presented in the submission appears unfair. Prior works such as LangSplat and OpenGaussian primarily focus on designing better mechanisms for embedding CLIP features into 3D Gaussians. For example, LangSplat employs an autoencoder to compress 512-D CLIP features, while OpenGaussian introduces a codebook-based embedding mechanism.

In contrast, the proposed method introduces a sparsification strategy that prunes redundant Gaussians for the target task. While this may indeed be a novel contribution, the paper lacks fair baseline comparisons to isolate the benefit of this sparsification. Specifically, a stronger evaluation would include variants such as 3DGS + existing Gaussian pruning methods + LangSplat/OpenGaussian, since prior works do not incorporate pruning for open-vocabulary 3D scene understanding. Without such baselines, it is difficult to attribute performance gains solely to the unified optimization framework.

This issue becomes more critical because the abstract explicitly highlights model size reduction and memory efficiency as key contributions. Therefore, comparisons with fair pruning-augmented baselines are essential.

__W2 Why unified / single optimization is necessary?__
While the paper claims novelty by unifying semantic learning, sparsification, and vector quantization into a single optimization step, the motivation for this unification is unclear.

_W2-1 What is the benefit of co-optimization?_
If the unified optimization substantially reduces overall training time or improves convergence efficiency, such results should be clearly reported. However, no experiments demonstrate improvements in training speed or optimization stability.

Furthermore, one could sequentially apply pruning and quantization to existing methods such as LangSplat or OpenGaussian. This sequential approach would also achieve model-size and memory improvements, raising the question: Why is a unified optimization scheme fundamentally better?
A direct comparison between: __Unified optimization (proposed) vs Sequential pipeline (prior methods + pruning/quantization)__ would help justify the methodological design. In addition to empirical results, a theoretical explanation supporting the necessity of unification would strengthen the contribution.

_W-2 Why optimize Gaussian parameters from scratch?_
Unlike previous language-3DGS works that rely on pre-optimized 3DGS Gaussians (following Kerbl et al.), this paper optimizes Gaussian parameters from scratch, jointly minimizing rendering and semantic losses. This appears to be motivated by the need to prune Gaussians during training.

However, the sparsification update described in Section 4.2 can also be applied on top of pre-optimized Gaussians. If my understanding is incorrect, clarification is needed.

As written, the current setup mixes 3DGS training efficiency with open-vocabulary semantic alignment, complicating direct comparison with existing methods. It would be more convincing if the method were also evaluated starting from pre-optimized 3DGS, consistent with LangSplat and OpenGaussian.

__W3 Missing details and insufficient analysis__
The paper states in line 191 that an autoencoder is used to compress 512-D CLIP embeddings, similar to LangSplat. However, the dimensionality of the compressed embeddings is not provided. This information is critical for understanding memory savings and representation quality.

Additionally, the proposed vector quantization strategy resembles the codebook training in OpenGaussian. Given that both methods employ similar training objectives for quantization/codebook learning, it remains unclear whether the performance improvements stem primarily from the unified optimization framework or other factors. More ablation or diagnostic experiments—particularly comparing the codebook quality between this paper and OpenGaussian—would help clarify this point.

__W4. Misleading statement__
In Lines 19-21, _"However, all the existing approaches are not designed for compressing language 3DGS, where rich semantic features are ignored during the compression stages ..."

However, in the recent paper, Dr.Splat [B], this paper does not compress, but keep the 512-D CLIP embeddings by Product Quantization. It is more closer to _quantization_ rather than to _compression_. Accordingly, __not__ all the existing methods compress the CLIP embeddings for this task which contradicts with the statements in Line19-21.

**Questions:**

__Q1. Training time comparison will be helpful.__

__Q2. learnable language vector __f__ or indices of codebook vector?__
In Line 232 of the manuscript, the proposed method also introduce the Gaussian parameter theta that involves the learnable language vector. If so, I am quite confused with the statement in Line 259, _"The unified optimization objective in Eq. 5 involves two non-differentiable components ... the penalty itself is differentiable but the discrete assignment is not"_

If the method assigns the index of the codebook vector on top of 3D Gaussians, it is true that the discrete assignment is not differentiable as stated in Line 261. However, I am not sure why the authors also introduce the language vector __f__? I personally expect that __f__ is not used, but I would like to double-check the implementation details.

+ it will be appreciated to revise Lines 230-238 to clearly describe the parametrization details.

__Q3. Does the authors also use autoencoder?__
In Line 189, _"LangSplat (Qin et al., 2024) employs an autoencoder to map CLIP embeddings
into a compact latent space for efficient rendering and reconstructs the full features when needed.
Our approach also adopts both strategies."_

However, in the methodology section, I cannot find where the autoencoder is used. When optimizing the codebook, does the authors follow the Vector Quantized Variational Autoencoder (VQ-VAE) [A] ? It is highly ambiguous how the training procedures are designed.

__Q-others. Please refer to the weakness sections and I hope to read the authors' responses.__

__Review summary__
The paper proposes a unified framework combining semantic learning, sparsification, and vector quantization for compact language-embedded 3DGS, but several concerns remain. First, the comparisons are not fair, as prior works like LangSplat and OpenGaussian do not incorporate pruning; stronger baselines combining them with existing pruning methods are needed, especially given the paper’s emphasis on model-size reduction. Second, the motivation for unifying all optimization steps is unclear, and the paper does not demonstrate why this co-optimization is superior to a sequential pipeline or why re-optimizing Gaussian parameters from scratch is necessary instead of starting from pre-optimized 3DGS as prior work does. Finally, several technical details are missing—such as the dimensionality of the compressed CLIP features—and it remains unclear whether performance gains stem from the unified optimization or simply from codebook-related improvements similar to OpenGaussian.

__References__
[A] Neural Discrete Representation Learning, Neurips 2017
[B] Dr.Splat, CVPR 2025

---

### Official Review · Reviewer_LpAn · 2025-11-01

**Soundness:** 3
**Presentation:** 4
**Contribution:** 4
**Rating:** 6
**Confidence:** 5

**Summary:**

This paper proposes CoLaSplat (Compact Language 3D Gaussian Splatting), a unified framework for compressing language-embedded 3D Gaussian Splatting (3DGS) models used in open-vocabulary 3D scene understanding.
Unlike existing sequential pruning and quantization methods, CoLaSplat integrates semantic learning, sparsification, and vector quantization into a single optimization problem solved by a primal-dual optimization scheme.
This unified approach jointly optimizes Gaussian parameters to preserve semantic and rendering fidelity while drastically reducing redundancy.

CoLaSplat achieves smaller model size and lower GPU memory, while maintaining or improving segmentation and localization accuracy on 3D-OVS and LERF benchmarks.

The paper also provides theoretical convergence guarantees and open-sources code and data to support reproducibility.

This reviewer looks forward to the authors’ rebuttal and revision responses and would be glad to reassess the rating after the discussion.

**Strengths:**

- Novel unified semantic 3DGS optimization: First method to jointly optimize compression and semantic alignment for language-embedded 3DGS, bridging a key efficiency–semantics gap.

- Strong efficiency-performance balance: Delivers significant compression without degrading visual or semantic quality, validated across multiple datasets and metrics.

- Reasonable evaluation: Benchmarked against five strong baselines with clear gains in mIoU, memory, and FPS, and supported by ablations on pruning and quantization effects.

- Reproducibility: Public code, datasets, and detailed algorithmic descriptions (Algorithm 1, Appendix C) enhance transparency and replicability.

**Weaknesses:**

- Practical complexity: The proposed optimization framework introduces additional computational overhead. However, the paper lacks an analysis or comparison of training time and overall computational cost.

- Overclaim: In the abstract and introduction, the authors claim a 147× faster inference. Where does this number come from? Compared to which baseline?

- FPS may be misleading 1 (Tables 1 and 2): Is the rendering speed comparison based on RGB images or feature maps? Additionally, did the authors apply any extra optimizations, such as CUDA acceleration? This reviewer finds Table 4 confusing, where even without pruning, the FPS (262) remains nearly unchanged. Where, then, does the reported speed gain originate? Vector quantization only affects to the memory or storage, not the computational time, right?

- FPS may be misleading 2 (Tables 1 and 2): The term "3D semantic segmentation" in LangSplat [Qin et al.] does not actually refer to 3D segmentation; instead, the evaluation measures 2D mask accuracy projected from 3D relevance.
In contrast, OpenGaussian operates in a fundamentally different regime. It directly queries and localizes 3D volumes without rendering.
Moreover, since OpenGaussian performs direct 3D-space queries without 2D rendering, as discussed in [C1], the overall computational complexity and cost of the entire system are significantly lower than those of any rendering-based approaches (including Feature-3DGS, GS-Grouping, LEGaussian, and LangSplat).
Therefore, comparing FPS results in Tables 1 and 2 with those rendering-based methods is inappropriate.

[C1] Dr. Splat: Directly Referring 3D Gaussian Splatting via Direct Language Embedding Registration, CVPR 2025.

- Missing citation and comparison: OpenGaussian and [C1] belong to a different methodological category from other competing approaches. Reference [C1] is missing in the citations.
The related work section should clearly distinguish these approaches from others including the proposed method. This distinction should also be reflected in the experimental comparisons.

- Missing Q as a function argument in Eq. (7) and Appendix C: This issue is critical. Up to Eq. (6),
𝑄 is treated as an optimization variable under the minimization operator, but it is suddenly omitted in Eq. (7) and in Appendix C. This is not a mere typographical error. The optimization process over
𝑄 must be considered in the proof. Consider revising the proof accordingly or omitting the proof entirely. Even without the theoretical proof, the contribution remains substantial (especially since convergence results for non-convex and non-smooth augmented Lagrangian multiplier methods up to a critical point are already well-established and do not constitute a novel contribution).

- Ablation breadth:The ablation studies are limited to pruning and quantization removal. Additional empirical analyses on parameter sensitivity or convergence robustness would further substantiate the claims. Although a theoretical convergence analysis is provided, corresponding empirical validation should also be included.

- No semantic degradation analysis: The paper does not analyze subtle semantic drift due to quantization, relying solely on global mIoU metrics.

**Questions:**

Please check the weakness section.

---

### Note · Authors · 2025-11-14

**Comment:**

I have read and agree with the venue's withdrawal policy on behalf of myself and my co-authors.

**Withdrawal Confirmation:**

I have read and agree with the venue's withdrawal policy on behalf of myself and my co-authors.